nanotechnology/inorganic chemistry/materials science

moisture-proof, mould-proof, LDH nanosheet, wood preservation, soundboards

**Authors for correspondence:**
Xingyun Li
e-mail: lixingyun@stu.xmu.edu.cn
Wanli Li
e-mail: liwanli@stu.xmu.edu.cn

This article has been edited by the Royal Society of Chemistry, including the commissioning, peer review process and editorial aspects up to the point of acceptance.

# Moisture- and mould-proof characteristics of surface modified wood for musical instrument soundboards

Caiping Xu[1], Zhenbo Liu[3], Xingyun Li[2] and Wanli Li[2]

[1]Department of Music, College of Art, and [2]Department of Biomaterials, College of Materials, Xiamen University, Xiamen 361005, People's Republic of China
[3]Key Laboratory of Bio-Based Material Science and Technology of the National Ministry of Education, Northeast Forestry University, 26 Hexing Road, Harbin 150040, Heilongjiang, People's Republic of China

XL, 0000-0002-8449-3767

Wood is the main material used for musical instrument soundboard fabrication, for practical and cultural reasons. As a natural material, however, wood is easily degraded due to moisture or fungal corrosion. Most traditional wood protection methods were devised for structural materials, and may thus not be suitable for application in musical instrument soundboard materials. In the current study, a novel nanomaterial-based modification method was applied to wood. The surface of wood was coated with polyurethane and MgAl-layered double hydroxide nanosheets after a convenient impregnation process. The modified wood exhibited improved hydrophobicity and mould-resistance, while maintaining its acoustic properties. This modified wood may facilitate the construction of soundboards with longer lifespans.

## 1. Introduction

Wood is the main processed material used to construct soundboards for various musical instruments, including pianos, violins, guitars and pipas [1]. The wood materials most commonly used in instrument construction have properties such as aesthetic appearance, comfortable tactile qualities, easy workability and suitable acoustic characteristics [2]. Accordingly, appropriate timbers for soundboards are usually expensive. However, as a biomaterial, wood is moisture-sensitive and has low resistance to fungi and insects. These factors can cause severe loss of density and strength, damaging the crucial vibrational properties of soundboards. In order to lower

the costs of the preservation and repair of wooden components of instruments, the development of effective strategies to protect and prolong the lifespan of wooden soundboards is of great importance [3–5].

Conventional wood protection strategies include thermal treatment, plasma treatment and chemical modification [6,7]. Thermal treatment is a very eco-friendly wood modification strategy because it does not generate pollutants [8,9]. The method has been commercialized for several years and can effectively enhance the durability of wood. The main drawback of thermally treating wood is that it degrades some of its mechanical properties [10]. Wood also becomes darker in colour after thermal treatment [8]. Another environmentally friendly modification method is plasma treatment. The surface hydrophobicity of wood can be increased via plasma treatment, without changing the inherent properties of the wood [11,12]. This technique has been practised for decades but the relevant mechanisms involved are not fully understood, making it hard to determine key experimental parameters. Plasma treatment can also yield inconsistent results [13]. Chemical modifications such as acetylation and furfurylation can increase the resistance of wood to moisture and fungi effectively without substantial loss of mechanical strength, but most chemical treatment methods generate polluting by-products and are only suitable for certain species of wood. Therefore, the development of a green, efficient and convenient strategy for the modification of wood used for musical instrument soundboards is required. Fortunately, with the development of nanoscience and nanotechnology in recent years, protecting wood with nanomaterials has become an option [14–16]. Wood cell walls exhibit porosity on a micrometre scale, which greatly facilitates the penetration of nanomaterials, resulting in the alteration of wood surface chemistry and improvement of the wood's properties. The nanomaterials can be used as coatings, or as polymeric nanocarriers via the impregnation of nano-based materials. Though the application of nanotechnology in wood preservation theoretically has great potential, associated research has remained at a developmental stage. Only a few kinds of nanoparticles have been used, including those made from copper, silver, zinc oxide, titanium dioxide and silicon dioxide [17]. Metal/metal oxide nanoparticles usually have highly active surfaces, and tend to aggregate into micrometer-scale clusters, increasing the difficulty of dispersion and the risk of detachment [18].

Compared to the frequently used nanoparticles, two-dimensional nanomaterials are more suitable for surface modification. The two-dimensional structure of these kinds of materials results in a large specific area, which can greatly reduce consumption of the coverage materials and increase the physical shield effect. MgAl-layered double hydroxide (LDH) is an ionic clay material with a unique layered structure that can be exfoliated to a two-dimensional nanosheet [19–21]. The well-known properties of MgAl-LDH include low cost, a good environmental profile, good anticorrosive properties and high anion exchange capacity. By changing the anions, the chemical composition of LDH can be flexibly adjusted to modify its characteristics. The rich hydroxyl groups on the LDH surface facilitate easy binding to biomaterials. Therefore, LDH nanosheets are promising materials for wood modification.

Polyurethane (PU) is widely used as a wood-coating polymer due to its transparence, ease of construction and strong adhesion properties [22,23]. LDH nanosheets are good polymer fillers for strengthening the network structure of wood, and preventing the sliding of molecular chains [24]. In the current study, a novel wood protection method was designed for wood used for musical instrument soundboards. MgAl-LDH nanosheets and waterborne PU were used. Co-modification with PU and LDH reduced the susceptibility of wood to moisture and mould, and the modification process had little effect on the acoustic parameters of the original wood, guaranteeing good vibro-mechanical properties of the wood for use in the construction of musical instrument soundboards.

# 2. Methods

## 2.1. Materials

Analytically pure magnesium nitrate hexahydrate [$Mg(NO_3)_2 \cdot 6H_2O$], aluminium nitrate nonahydrate [$Al(NO_3)_3 \cdot 9H_2O$], hexamethyleneimine, *N,N*-dimethylformamide and formamide were purchased from Xilong Scientific Co., Ltd. (Guangdong, China). Waterborne PU was purchased from Guangzhou Guanzhi New Material Technology (solid content 30%). 1H, 1H, 2H, 2H-perfluorodecyltrimethoxysilane (PFDTS) was purchased from Alfa (Shanghai, China). Mucor powder was purchased from Yishui Jinrun Biotechnology Co., Ltd.

## 2.2. LDH synthesis

LDH was synthesized as previously described [25]. First, 0.02 mol $Mg(NO_3)_2 \cdot 6H_2O$, 0.01 mol $Al(NO_3)_2 \cdot 9H_2O$ and 0.026 mol hexamethyleneimine were mixed thoroughly in 80 ml distilled water. The mixture was then transferred to an autoclave for a hydrothermal reaction (140°C, 24 h). The MgAl-$CO_3$ obtained was washed with water and ethanol several times, then dried in a vacuum drying oven at 40°C, after which 0.5 g MgAl-$CO_3$ was dispersed into 500 ml distilled water with 0.75 mol $NaNO_3$ and 0.0025 mol $HNO_3$. The dispersion solution was tightly sealed after purging with argon gas, and was shaken for 24 h at ambient temperature. After the ion-exchange process, the MgAl-$NO_3$ obtained was washed with water and ethanol several times, then dried in a vacuum drying oven. To exfoliate the LDH into nanosheets, 0.05 g of the MgAl-$NO_3$ LDH powder obtained was mixed with 100 ml formamide in a flask, which was tightly sealed after purging with argon gas. The mixture was then vigorously stirred for 48 h.

## 2.3. Modification of wood

The wood used in the current study was *Paulownia elonata*, which is commonly used for the construction of musical instrument soundboards. All wood specimens used were $2.0 \times 2.0 \times 1.0$ cm, except those used for the characterization of acoustic properties, and the average density of the wood was 0.236 g cm$^{-3}$. All specimens were pre-dried in a vacuum oven at 60°C overnight, then fully immersed in mixtures of PU and different concentrations of LDH for 24 h. Based on LDH content, the samples were labelled PU (no LDH), PU/LDH-0.5 (0.5% weight LDH), PU/LDH-1, PU/LDH-2, PU/LDH-4 and PU/LDH-6. After impregnation, the specimens were put on a silica gel plate and left to air-dry. To ensure complete drying, the specimens were then placed in a vacuum drying oven at 60°C for 2 h then 120°C for 5 h.

Simplified thermal treatment and sol-gel treatment methods were used to generate additional control wood samples. The thermal treatment process was performed by putting wood samples in a closed teflon reactor filled with steam (190°C, 5 h). The sol-gel treatment was performed by putting wood samples into 1 mM PFDTS/*N,N*-dimethylformamide solution for 24 h. These two types of samples were subsequently dried in a vacuum drying oven at 60°C for 2 h then at 120°C for 5 h.

## 2.4. Material characterizations

Micrographs of samples (LDH sheets and wood) were generated via scanning electron microscopy (SEM; SU70, Hitachi). SEM images were acquired by an electron backscatter diffraction detector at 10 kV with a working distance of 10 mm. For LDH characterization, the samples were observed directly by SEM without extra treatment. All wood samples were dried at 100°C for another 5 h, then sputtered with a thin layer of Au via a vacuum evaporation coating apparatus (ETD-800C, Beijing Boyuan Micro Nano Technology Co. Ltd.). Energy dispersive spectroscopy characterization was conducted using an Oxford Aztec EDX instrument equipped in the SU-70 SEM. The surfaces of wood samples were also observed via laser-scanning microscopy (LSM, VK-X200, Keyence) without any pretreatment.

Fourier-transform infrared spectrometry (FTIR) was measured using NICOLET iS10 at a resolution of 4 cm$^{-1}$ and 32 scans in the range from 700 cm$^{-1}$ to 4000 cm$^{-1}$. The testing samples were obtained by cutting the surfaces of various wood samples, then grinding to make small chips. Three different chips from the same wood sample were analysed to ensure that the spectra obtained represented the whole board.

Atomic force microscopy images were recorded using an atomic force microscope (DI Multimode, Veeco) in tapping mode. X-ray diffraction patterns were recorded using a Philips X'Pert Pro (Philips, Amsterdam, Netherlands; $\lambda = 1.54056$ Å) with Cu Ka radiation. The running parameters were kept at 40 kV, 35 mA. The resolution was 0.02°/s. Contact angle (CA) measurements were performed using the sessile drop method with a contact angle tester (DSA20, Kruss). The volume of the individual distilled water droplet in the static CA test was 5 µl. Drop shape was recorded with a digital camera after 5 s. CA values were calculated from 20 separate measurements of each sample. Specifically, each wood block had five test points, and each type of modified wood had four parallel test samples.

## 2.5. Water absorption tests

Water absorption (WA) values were determined via a water-soaking method. The specimens were weighed, then submerged in distilled water at room temperature. At set intervals, the specimens were taken out, surface water was wiped off, then they were weighed again. Triplicate specimens were assessed for each

condition. WAs were determined using the following equation, where $M_1$ is the weight of the specimen before tests and $M_2$ is the weight of the water-saturated specimen

$$WA(\%) = 100 \times \frac{M_2 - M_1}{M_1}.$$

## 2.6. Mould resistance

To simulate fungal corrosion conditions in practical applications, wood specimens were kept in a damp, hot, dark environment (humidity >85%, temperature approx. 30°C) for five weeks with mucor powder. Mould resistance was also assessed in specimens after immersion in water for 24 h (leaching). After incubation, the samples were washed with water and wiped with a clean cloth. They were then dried in a vacuum drying oven at 60°C for 24 h followed by 120°C for 2 h. The dried samples were then weighed. Triplicate specimens were used for each condition.

## 2.7. Acoustic properties

Acoustic properties were assessed using a dual-channel fast Fourier-transform analyzer (CF-5220Z, Ono Sokki, Japan). The wood samples used in the tests were $30 \times 3 \times 1$ cm in size. Each wood sample was placed between two foam supports (nodal positions) with its two ends suspended in the air to facilitate free vibration. Nodal positions were set at 0.224 of the total length from each end of the sample. A small piece of hardwood with high density was then used to knock one end of the sample. A high-sensitivity sensor was linked to the other end to record acoustic vibrations. When measuring surface wave propagation speed, the sensor was linked to both ends of the sample. During the tests five measurements were taken for each board. The data collected (sound signals) were amplified, filtered and assessed using the fast Fourier-transform analyser to obtain resonance frequencies. The dynamic elastic modulus ($E/\rho$), acoustic radiation damping coefficient ($R$) and acoustic impedance ($\omega$) were calculated. The equation used to calculate the dynamic elastic modulus was as follows [26], where $L$ is the board length, $f$ is the resonance frequency, $\beta_n$ is the relative constant of wood boundary conditions and $h$ is the board thickness

$$\frac{E}{\rho} = \frac{48\pi^2 L^4 f^2}{\beta_n^4 h^2}.$$

The equation used to calculate $R$ was

$$R = \frac{v}{\rho} = \sqrt{\frac{E}{\rho^3}}.$$

The equation used to calculate $\omega$ was

$$\omega = \rho v = \sqrt{\rho E}.$$

# 3. Results and discussion

## 3.1. LDH characterization

MgAl-CO$_3$ LDHs were synthesized using a conventional hydrothermal reaction method. Figure 1$a$ is an SEM image of the MgAl-CO$_3$ LDHs obtained, depicting a complete hexagon nanoplate morphology. Hydrated anions located in the interlayer region of LDHs were then changed from $CO_3^{2-}$ to $NO_3^-$ via the ion exchange method. The MgAl-NO$_3$ LDHs exhibited similar nanoplate morphology to the MgAl-CO$_3$ LDHs but with larger interlamellar spacing (figure 1$b$), which can favour the exfoliation process. An SEM image of the final exfoliated LDH nanosheets with sizes of approximately 1.6–4.2 µm is shown in figure 1$c$. The thickness of the nanosheets was determined via atomic force microscopy (figure 1$d$), and ranged from 0.8–2.6 nm (approx. 1–3 layers). LDH dispersion exhibited a clear Tyndall effect (figure 1$e$), indicating that the nanosheets existed as nearly colourless colloids in aqueous solution. This suggested that the LDH coating on the surface of the wood would not change its colour.

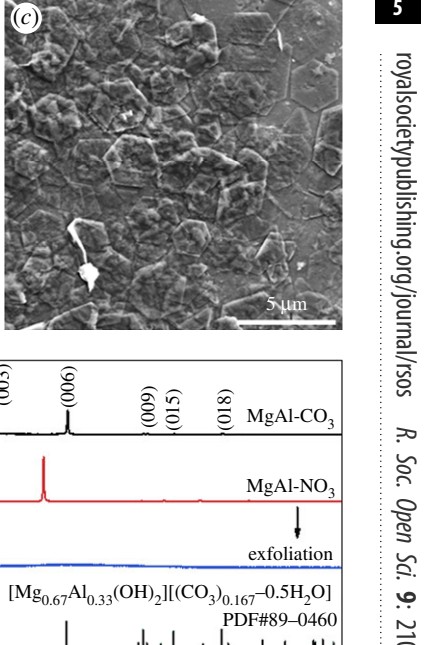

**Figure 1.** SEM images of (*a*) MgAl-CO$_3$, (*b*) MgAl-NO$_3$ and (*c*) exfoliated MgAl-LDH. (*d*) Atomic force microscopy image and (*e*) optical image of colloidal exfoliated MgAl-LDH solution. Tyndall effect was visible when irradiated with a laser beam. (*f*) X-ray diffraction patterns of MgAl-CO$_3$, MgAl-NO$_3$ and exfoliated MgAl-LDH.

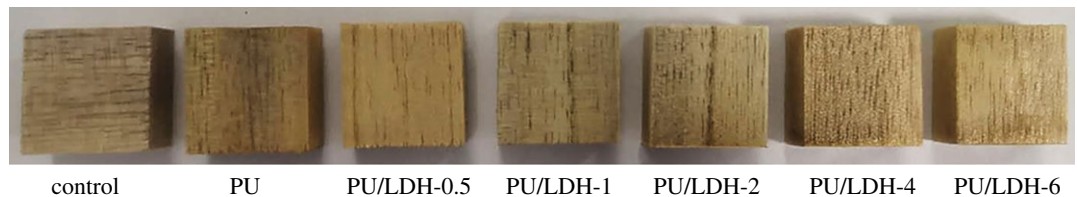

**Figure 2.** A photograph of wood specimens with and without PU/LDH modification.

MgAl-CO$_3$ and MgAl-NO$_3$ LDH X-ray diffraction results are shown in figure 1*f*. MgAl-NO$_3$ LDH peaks were located at lower angle position compared with the MgAl-CO$_3$ precursor, confirming layer spacing enlargement [27]. No obvious peaks in the X-ray diffraction pattern of the LDH nanosheets were observed, due to the disordered arrangement of the exfoliated laminates [25]. All the above results indicated the successful synthesis and exfoliation of high-quality MgAl-LDH.

## 3.2. Characterizations of the modified wood

In experiments using different ratios of PU/LDH during the impregnation process, the PU polymer was translucent, and the MgAl-LDH dispersion was an almost colourless emulsion (figure 1*e*). Accordingly, the PU/LDH modification did not change the original colour of the wood (figure 2). There were no obvious changes in the appearance of the wood associated with increased LDH content.

In Fourier-transform infrared (FTIR) spectroscopy characterization peaks located at approximately 1235 cm$^{-1}$ (C–O group), 1725 cm$^{-1}$ (C=O group), 2800–2900 cm$^{-1}$ (C–H group) and 3315 cm$^{-1}$ (N-H group) can be ascribed to the characteristic absorption peaks of PU (electronic supplementary material, figure S1a) [28]. Characteristic LDH peaks were not obvious in FTIR spectra however (electronic supplementary material, figure S1b), and the complex surface chemistry of wood also impeded the verification of characteristic LDH peaks via FTIR.

The micro-morphologies of the wood specimens were assessed via LSM and SEM, and because of the anisotropy of the structure of wood the morphologies of the transverse surfaces and longitudinal surfaces of the specimens were investigated separately. Electronic supplementary material, figure S2 is a typical

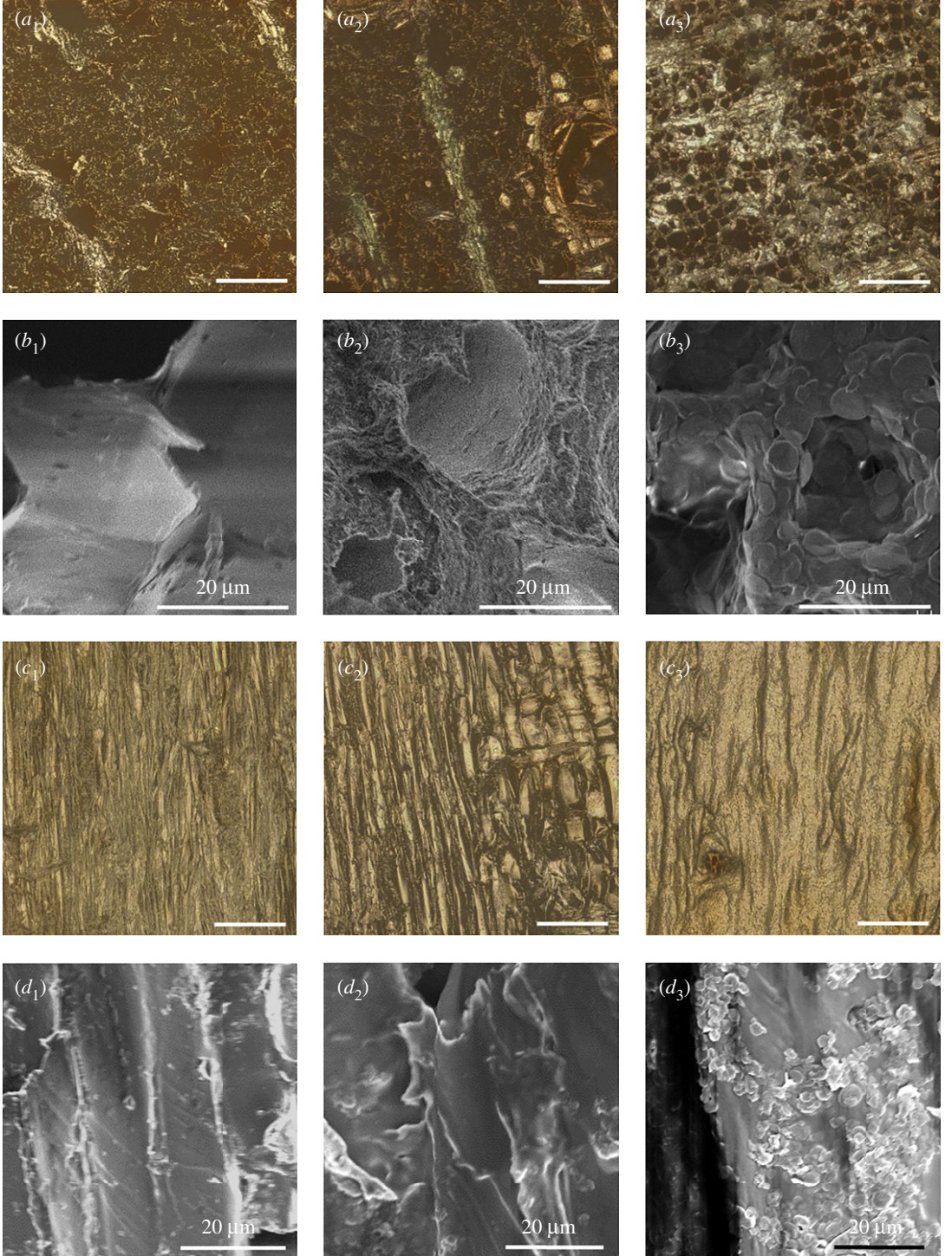

**Figure 3.** Row (*a*) LSM images of transverse surfaces. (*b*) SEM images of transverse surfaces. (*c*) LSM images of longitudinal surfaces. (*d*) SEM images of longitudinal surfaces. Column 1: Control (original wood). Column 2: PU (PU coated wood). Column 3: PU/LDH-6 (PU and 6% weight LDH-coated wood). Each scale bar in LSM images: 200 µm.

SEM image of the transverse surface of wood, depicting micropores with sizes ranging from 8 to 50 µm. The average diameter of pores in the wood cell wall was approximately 30 µm. Figure $3a_1$–$a_3$ is the respective LSM images of the transverse surfaces of original wood, PU-coated wood and a PU/LDH-6 sample. The porous morphology of wood did not exhibit any change after PU or PU/LDH modifications, but due to the PU coating the modified wood exhibited an adhesive layer in SEM images (figure $3b_1$, $b_2$). LDH nanosheets were clearly evident in SEM images, and they also infiltrated the pores of wood (figure $3b_3$). Unlike the transverse surfaces, the longitudinal surfaces of wood present an aligned fibre-like morphology, as depicted in LSM images (figure $3c_1$–$c_3$). Coating with PU/LDH preparations with

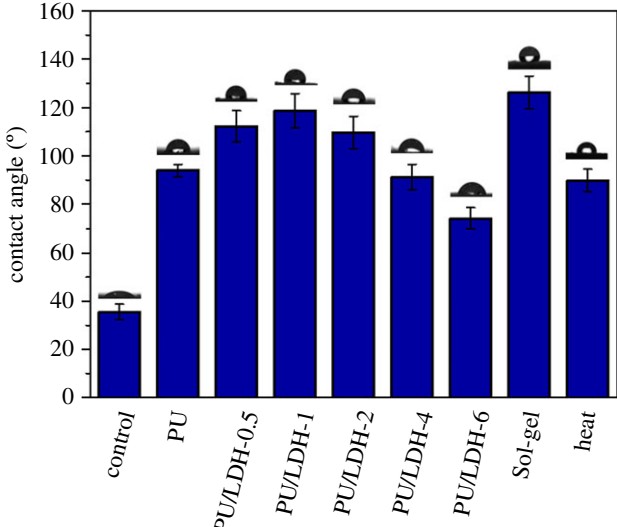

**Figure 4.** CAs of different wood specimens.

different LDH contents did not induce morphology changes on the longitudinal surfaces. In SEM imaging, the existence of PU was not as clear as it was in transverse surface images, whereas LDH was clearly visible (figure $3d_1$–$d_3$). Energy dispersive spectroscopy results and elements distribution mapping images clearly indicated the existence of Mg and Al elements in both transverse and longitudinal surfaces of the PU/LDH-1 sample (electronic supplementary material, figure S3–S5).

## 3.3. Properties of modified wood

### 3.3.1. Moisture-proofing

Water repellency was tested via dynamic CA measurements with distilled water. Water CA values of PFDTS sol-gel-treated and heat-treated wood specimens are shown in figure 4. These two different modification approaches were used to prepare specimens because PFDTS sol-gel treatment is an effective hydrophobic modification method and heat treatment is a conventional wood modification method. The data used to compile figure 4 are shown in electronic supplementary material, table S1. After PU modification the CA of the wood surface was significantly increased, from $35.5° \pm 3.3°$ to $93.9° \pm 2.5°$. The high amount of −OH on the surface of LDH ordinarily confers hydrophilicity. In the current study however, as the amount of LDH increased, the CA first increased slightly then decreased. This may have been due to the weak interaction force (electrostatic repulsion) between the positively charged LDH and the anionic PU. When the LDH content was small the bridging between LDH and PU could benefit the coating by forming a spatial network structure [24]. Further increasing the LDH content caused more and more −OH to be exposed on the coating surface, and eventually, the hydrophilicity of the sample was enhanced. PFDTS sol-gel treatment is used to generate superhydrophobic surfaces, and concordantly in the current study the CA of PFDTS-treated samples increased a lot ($126.2° \pm 6.8°$). Heat treatment also increased the CA of wood to $89.8° \pm 4.6°$. Compared with the two conventional methods, the hydrophobization effect of PU/LDH modification was also sufficient (optimum = $118.7° \pm 7.1°$).

The retention levels of the modification layers and the leaching of PU/LDH from wood specimens were assessed to evaluate the lifespan of the coating. The leaching rates (weight percentage variation) of the different coatings after a 24-h leaching process (soaking in water) are shown in figure 5a. The leaching rate of pure PU coating was up to 72.3%. In the current study, the wood specimens were only modified via a simple impregnation method without further curing processes, therefore this leaching rate of the waterborne PU coating layer is reasonable [22]. However, with increased LDH content (≤1.0 weight percentage) in the PU/LDH coating, the leaching rate was effectively reduced. The PU/LDH-1 sample exhibited the smallest leaching rate among all samples, which is consistent with the water repellence results. A possible reason for reduced leeching of coating containing LDH is that the two-dimensional nanostructure of LDH can improve the adhesion ability of the coating [29]. Increased LDH content in the PU coating can decrease the CA due to a high amount of −OH on the surface of LDH, and thus leads to a higher leaching rate. The WA values of unmodified wood, PU-modified wood and PU/LDH-modified

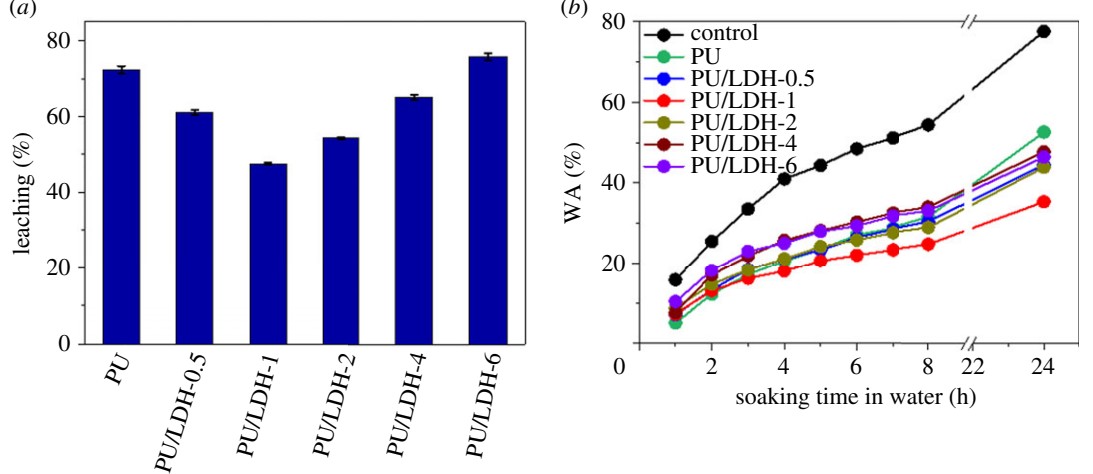

**Figure 5.** (*a*) Leaching rates of the wood specimens, (*b*) WAs of the wood specimens.

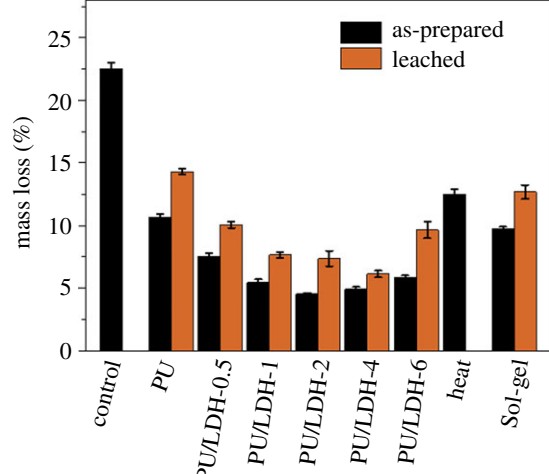

**Figure 6.** Weight loss of unmodified wood (control) and woods subjected to different modifications before and after leaching treatment.

wood are shown in figure 5*b* and electronic supplementary material, table S2. Because the weight of the coating was much smaller than that of the absorbed water, weight loss from leaching was negligible. Unmodified wood specimens exhibited WA values of nearly 80% in 24 h, indicating poor moisture resistance. After modification with PU or PU/LDH coatings, WAs were significantly reduced. Among all the wood specimens, the PU/LDH-1 sample exhibited the lowest WA of 32% in 24 h, indicating the best moisture proofing, which was probably due to enhanced hydrophobicity.

### 3.3.2. Mould-proofing

Mould repellency test results are shown in electronic supplementary material, figure S6. Because the main form of damage done to wood by mould is weight loss, mould-proofing is inversely proportional to the change in mass [17]. PU-modified wood exhibited superior mould-proofing as indicated by approximately 10.7% weight loss, which was much lower than the 22.5% weight loss exhibited by unmodified wood (figure 6*a*; electronic supplementary material, table S3). PU/LDH-modified woods exhibited much lower weight loss than PU-modified wood, indicating greater mould-proofing. Among all the PU/LDH-modified woods the PU/LDH-1 sample exhibited the lowest weight loss of 7.7%, which can probably be attributed to the best water repellence. All PU/LDH-modified woods exhibited better mould-proofing than PU-modified wood, even after the leaching treatment process. PFDTS sol-gel-treated wood exhibited the highest CA, but its mould-proofing was inferior to that of PU/LDH-modified wood. Therefore, the antibiosis properties of metal ions can also contribute to the mould resistance of PU/LDH-modified wood.

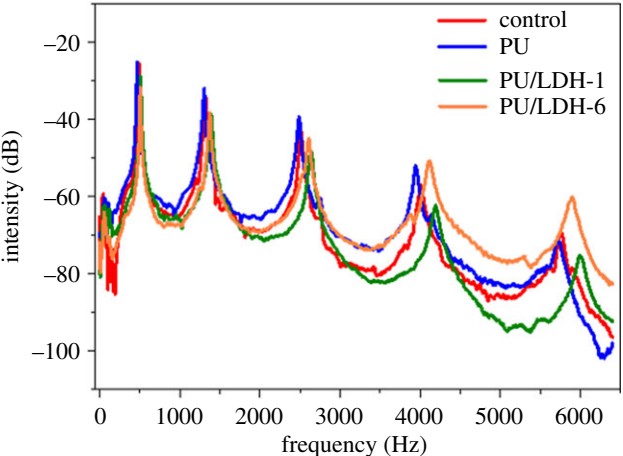

**Figure 7.** Vibration frequency spectra of different soundboards.

**Table 1.** Density values and acoustic parameters of different wood samples.

|          | $\rho$ (g cm$^{-3}$) | $E/\rho$ | $R$ | $\omega$ |
|----------|----------------------|----------|-----|----------|
| control  | 0.28 | 20.10 ± 2.11 | 15.77 | 1.27 |
| PU       | 0.29 | 20.44 ± 2.36 | 15.54 | 1.31 |
| PU/LDH-1 | 0.31 | 20.37 ± 2.14 | 14.77 | 1.38 |
| PU/LDH-6 | 0.30 | 19.71 ± 2.13 | 14.57 | 1.35 |

## 3.4. Acoustic properties

The acoustic properties of wood used for soundboards are a very important determinant of the quality of musical instruments. It is desirable that any soundboard modification not affect its acoustic properties. Generally, the most important acoustic properties of wooden soundboards are the specific dynamic $E/\rho$, $R$ and $\omega$, which were measured via the above-described methods in the current study [30,31]. The results are shown in table 1.

The $\rho$ values of the modified wood samples were only slightly higher than those of unmodified wood, indicating that the PU and PU/LDH coating layers on the wood surface were thin. The $E/\rho$, $R$ and $\omega$ values of different wood samples were also similar. The inherent vibration frequency spectrum signals of unmodified *P. elonata* (control) and the modified samples are shown in figure 7. There were five clear vibration frequencies corresponding to the five main orders in the vibration mode of the *P. elonata* wood, representing its vibration performance [32]. These five frequencies could be observed in the spectra of all samples, with similar peak shapes. Because PU/LDH coating is mainly a surface-related process and the thickness of the PU/LDH coating is only approximately 2.8 μm, the PU/LDH modification does not change the overall density or the interior wood grain. Density and wood grain are crucial parameters with respect to acoustic properties. The PU/LDH modification did not substantially affect the acoustic properties of the treated wood soundboards.

## 4. Conclusion

PU and nanoscale MgAl-LDH wood modification can improve the mould-proof and moisture-proof properties of wood. The optimal LDH content was determined by evaluating wood samples modified with different PU/LDH ratios. When the LDH content was 1% the PU/LDH-coated wood had a hydrophobic surface with a CA of 118.7° ± 7.1°. The mass loss of the PU/LDH wood sample was smaller than that of control samples after the same mould growth process, verifying its enhanced mould resistance. Vibration tests indicated that the PU/LDH coating did not noticeably affect the acoustic properties of the wood. This novel protection strategy is a promising method for protecting wood materials used for musical instrument soundboards.

**Data accessibility.** The authors confirm that the additional data supporting the findings of this study, including FTIR spectra, SEM images (energy dispersive spectroscopy mapping) and tables of repeated test results, are available within the electronic supplementary material [33]. The raw results of acoustic tests, CA tests and LDH characterizations are also available from the Dryad Digital Repository: https://doi.org/10.5061/dryad.5x69p8d24 [34].

**Authors' contributions.** All authors gave final approval for publication and agreed to be held accountable for the work performed therein.

**Competing interests.** We declare we have no competing interests.

**Funding.** This study was financially supported by National Natural Science Foundation of China (grant no. 31670559).

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
