## [Peer Review File · Royal Society Open Science]

Review History

RSOS-210042.R0 (Original submission)

Review form: Reviewer 1

Is the manuscript scientifically sound in its present form?

No

Are the interpretations and conclusions justified by the results?

Yes

Is the language acceptable?

No

Do you have any ethical concerns with this paper?

No

Have you any concerns about statistical analyses in this paper?

Yes

Recommendation?

Reject

Comments to the Author(s)

Please find the comments, suggestions and questions about the manuscript in the attached file (Appendix A).

Review form: Reviewer 2**Is the manuscript scientifically sound in its present form?**

Yes

Are the interpretations and conclusions justified by the results?

Yes

Is the language acceptable?

Yes

Do you have any ethical concerns with this paper?

No

Have you any concerns about statistical analyses in this paper?

No

Recommendation?

Accept as is

Comments to the Author(s)

Caiping Xu et al. entitled "Moisture- and Mold-proof Characteristics of Surface Modified Wood for Musical Instrument Soundboards" has reported a novel nanomaterial-based modification method applied to effectively protect wood from moisture and fungi, without changing the acoustic properties of wood. Author has claimed that the life span of the soundboards is improved.

At a small doping amount, the LDH could effectively enhance the wood protection performance of PU coating, which was mainly contributed to the unique 2D nanostructure. The results are organized in systematic manner and effective characterization was done in order to draw the conclusion. Though the limited characterization for sound analysis, the vibration frequency spectra of different soundboards is sensible. The various coating and its comparison with nanoscale materials is interesting study for real life applications.

Decision letter (RSOS-210042.R0)

Dear Dr Li:

Manuscript ID: RSOS-210042

Title: "Moisture- and Mold-proof Characteristics of Surface Modified Wood for Musical Instrument Soundboards"

Thank you for submitting the above manuscript to Royal Society Open Science. Your paper was sent to reviewers and their comments are included at the bottom of this letter.

In view of the concerns raised by the reviewers, the manuscript has been rejected in its current form. However, a new manuscript may be submitted which takes into consideration these comments.

Please note that resubmitting your manuscript does not guarantee eventual acceptance, and that your resubmission will be subject to peer review before a decision is made.

Your resubmitted manuscript should be submitted by 29-Aug-2021. If you are unable to submit by this date please contact the Editorial Office.

On behalf of the Subject Editor Professor Anthony Stace and the Associate Editor Dr Dattatray Late

REVIEWER(S) REPORTS:

Associate Editor Comments to Author ():

RSC Associate Editor:

Comments to the Author:

(There are no comments.)

RSC Subject Editor:

Comments to the Author:

(There are no comments.)

Reviewers' Comments to Author:

Reviewer: 1

Comments to the Author(s)

Please find the comments, suggestions and questions about the manuscript in the attached file.

Reviewer: 2

Comments to the Author(s)

Caiping Xu et al. entitled "Moisture- and Mold-proof Characteristics of Surface Modified Wood for Musical Instrument Soundboards" has reported a novel nanomaterial-based modification method applied to effectively protect wood from moisture and fungi, without changing the acoustic properties of wood. Author has claimed that the life span of the soundboards is improved.

At a small doping amount, the LDH could effectively enhance the wood protection performance of PU coating, which was mainly contributed to the unique 2D nanostructure. The results are organized in systematic manner and effective characterization was done in order to draw the conclusion. Though the limited characterization for sound analysis, the vibration frequency spectra of different soundboards is sensible. The various coating and its comparison with nanoscale materials is interesting study for real life applications.

Author's Response to Decision Letter for (RSOS-210042.R0)

See Appendix B.

RSOS-210790.R0

Review form: Reviewer 3 (Mehran Roohnia)

Is the manuscript scientifically sound in its present form?

Yes

Are the interpretations and conclusions justified by the results?

Yes

Is the language acceptable?

Yes

Do you have any ethical concerns with this paper?

No

Have you any concerns about statistical analyses in this paper?

No

Recommendation?

Accept with minor revision (please list in comments)

Comments to the Author(s)

The only suggestion of mine to improve the literatures is adding a comprehensive reference and citation for the equations provided for the acoustic property tests. The authors forgot to cite a reference there!

I suggest:

<https://doi.org/10.1016/B978-0-12-803581-8.01996-2>

Review form: Reviewer 4

Is the manuscript scientifically sound in its present form?

Yes

Are the interpretations and conclusions justified by the results?

Yes

Is the language acceptable?

Yes

Do you have any ethical concerns with this paper?

No

Have you any concerns about statistical analyses in this paper?

Yes

Recommendation?

Reject

Comments to the Author(s)

The work is not novel and not upto the standards of the journal

Decision letter (RSOS-210790.R0)

Dear Dr Li:

Title: Moisture- and Mold-proof Characteristics of Surface Modified Wood for Musical Instrument Soundboards
Manuscript ID: RSOS-210790

The editor assigned to your paper has now received comments from reviewers. We would like you to revise your paper in accordance with the referee and Subject Editor suggestions which can be found below (not including confidential reports to the Editor). Please note this decision does not guarantee eventual acceptance.

Please submit a copy of your revised paper before 14-Jul-2021. Please note that the revision deadline will expire at 00.00am on this date. If we do not hear from you within this time then it will be assumed that the paper has been withdrawn. In exceptional circumstances, extensions may be possible if agreed with the Editorial Office in advance. We do not allow multiple rounds of revision so we urge you to make every effort to fully address all of the comments at this stage. If deemed necessary by the Editors, your manuscript will be sent back to one or more of the

original reviewers for assessment. If the original reviewers are not available we may invite new reviewers.

On behalf of the Subject Editor Professor Anthony Stace and the Associate Editor Dr Dattatray Late.

RSC Associate Editor
Comments to the Author:
Major Revision: Authors need to bring the novelty of this manuscript.

Reviewers' Comments to Author:

Reviewer: 3

Comments to the Author(s)

The only suggestion of mine to improve the literatures is adding a comprehensive reference and citation for the equations provided for the acoustic property tests. The authors forgot to cite a reference there!

I suggest:

<https://doi.org/10.1016/B978-0-12-803581-8.01996-2>

Reviewer: 4

Comments to the Author(s)

The work is not novel and not upto the standards of the journal

Author's Response to Decision Letter for (RSOS-210790.R0)

See Appendices C & D.

RSOS-210790.R1 (Revision)

Review form: Reviewer 4

Is the manuscript scientifically sound in its present form?

No

Are the interpretations and conclusions justified by the results?

No

Is the language acceptable?

No

Do you have any ethical concerns with this paper?

No

Have you any concerns about statistical analyses in this paper?

No

Recommendation?

Reject

Comments to the Author(s)

The paper was not improved

Review form: Reviewer 5

Is the manuscript scientifically sound in its present form?

Yes

Are the interpretations and conclusions justified by the results?

Yes

Is the language acceptable?

Yes

Do you have any ethical concerns with this paper?

No

Have you any concerns about statistical analyses in this paper?

No

Recommendation?

Accept as is

Comments to the Author(s)

Moisture-proof is very important for wood used as musical instrument soundboards. In this work, MgAl layered double hydroxide (MgAl-LDH) nanosheet as a cheap, chemical stable and environmentally friendly ion clay material was synthesized on the surfaces of wood. The co-modification by PU and LDH could effectively improve the moisture- and mold-proof properties of wood. Furthermore, the modification process had little effect on the acoustic parameters of the original wood, guaranteeing the great vibromechanical properties of wood as musical instrument soundboards. Since the novelty of this work has been addressed by the authors. The authors have also improved their manuscript according to the reviewers' comments. I would like to recommend its acceptance for publication.

Review form: Reviewer 6**Is the manuscript scientifically sound in its present form?**

Yes

Are the interpretations and conclusions justified by the results?

Yes

Is the language acceptable?

Yes

Do you have any ethical concerns with this paper?

No

Have you any concerns about statistical analyses in this paper?

No

Recommendation?

Accept with minor revision (please list in comments)

Comments to the Author(s)

The use of inappropriate verbs or prepositions may change the meaning of the sentence, hence better get the manuscript rechecked with some expert in English.

Although authors could discuss the conventional wood protection strategies, the limitations of the metal/metal oxide nanoparticles coating seem missing. Authors directly jump to LDH, leaving the link from nanoparticles(0D) to 2D nanomaterials missing.

The Moisture-proof property, as studied with contact angle measurement (Fig. 4) indicate PFDTs sol-gel treatment to be the best. In such a case, it is hard to understand the reason for skipping the PFDTs sol-gel treatment in some of the subsequent studies. Authors are requested to shed light on the same.

Decision letter (RSOS-210790.R1)

Dear Dr Li:

Title: Moisture- and Mold-proof Characteristics of Surface Modified Wood for Musical Instrument Soundboards
Manuscript ID: RSOS-210790.R1

Thank you for submitting the above manuscript to Royal Society Open Science. On behalf of the Editors and the Royal Society of Chemistry, I am pleased to inform you that your manuscript will be accepted for publication in Royal Society Open Science subject to minor revision in accordance with the referee suggestions. Please find the reviewers' comments at the end of this email.

The reviewers and handling editors have recommended publication, but also suggest some minor revisions to your manuscript. Therefore, I invite you to respond to the comments and revise your manuscript.

Please also include the following statements alongside the other end statements. As we cannot publish your manuscript without these end statements included, if you feel that a given heading is not relevant to your paper, please nevertheless include the heading and explicitly state that it is not relevant to your work. We have included a screenshot example of the end statements for reference.

- Ethics statement

Please clarify whether you received ethical approval from a local ethics committee to carry out your study. If so please include details of this, including the name of the committee that gave consent in a Research Ethics section after your main text. Please also clarify whether you received informed consent for the participants to participate in the study and state this in your Research Ethics section.

OR

Please clarify whether you obtained the necessary licences and approvals from your institutional animal ethics committee before conducting your research. Please provide details of these licences and approvals in an Animal Ethics section after your main text.

OR

Please clarify whether you obtained the appropriate permissions and licences to conduct the fieldwork detailed in your study. Please provide details of these in your methods section.

- Data accessibility

It is a condition of publication that you make available the data and research materials supporting the results in the article. Datasets should be deposited in an appropriate publicly available repository and details of the associated accession number, link or DOI to the datasets must be included in the Data Accessibility section of the article (<https://royalsocietypublishing.org/rsos/for-authors#question17>). Reference(s) to datasets should also be included in the reference list of the article with DOIs (where available).

Please include a Data Availability section after your main text stating where supporting data are available from, or where they will be made available should your article be accepted for publication.

If you wish to submit your supporting data or code to Dryad (<http://datadryad.org/>), or modify your current submission to dryad, please use the following link:
<http://datadryad.org/submit?journalID=RSOS&manu=RSOS-210790.R1>

- **Competing interests**

Please include a Competing Interests section after your main text declaring any financial or non-financial competing interests. If you have no competing interests please state 'I/we have no competing interests.'

- **Authors' contributions**

Please include an Authors' Contributions section at the end of your main text detailing the contribution of each author. All authors should have read and approved the manuscript before submission and this should be stated in the Authors' Contributions section.

The list of Authors should meet all of the following criteria; 1) substantial contributions to conception and design, or acquisition of data, or analysis and interpretation of data; 2) drafting the article or revising it critically for important intellectual content; and 3) final approval of the version to be published.

- **Acknowledgements**

- **Funding statement**

Please include a funding section after your main text which lists the source of funding for each author.

Because the schedule for publication is very tight, it is a condition of publication that you submit the revised version of your manuscript before 22-Oct-2021. Please note that the revision deadline will expire at 00.00am on this date. If you do not think you will be able to meet this date please let me know immediately.

Kind regards,
Dr Ellis Wilde
Publishing Editor, Journals

On behalf of the Subject Editor Professor Anthony Stace and the Associate Editor Dr Dattatray Late.

RSC Associate Editor
Comments to the Author:
Accept with minor revisions

RSC Subject Editor
Comments to the Author:
(There are no comments.)

Reviewer comments to Author:

Reviewer: 4

Comments to the Author(s)

The paper was not improved

Reviewer: 5

Comments to the Author(s)

Moisture-proof is very important for wood used as musical instrument soundboards. In this work, MgAl layered double hydroxide (MgAl-LDH) nanosheet as a cheap, chemical stable and environmentally friendly ion clay material was synthesized on the surfaces of wood. The co-modification by PU and LDH could effectively improve the moisture- and mold-proof properties of wood. Furthermore, the modification process had little effect on the acoustic parameters of the original wood, guaranteeing the great vibromechanical properties of wood as musical instrument soundboards. Since the novelty of this work has been addressed by the authors. The authors have also improved their manuscript according to the reviewers' comments. I would like to recommend its acceptance for publication.

Reviewer: 6

Comments to the Author(s)

The use of inappropriate verbs or prepositions may change the meaning of the sentence, hence better get the manuscript rechecked with some expert in English.

Although authors could discuss the conventional wood protection strategies, the limitations of the metal/metal oxide nanoparticles coating seem missing. Authors directly jump to LDH, leaving the link from nanoparticles(0D) to 2D nanomaterials missing.

The Moisture-proof property, as studied with contact angle measurement (Fig. 4) indicate PFDTS sol-gel treatment to be the best. In such a case, it is hard to understand the reason for skipping the PFDTS sol-gel treatment in some of the subsequent studies. Authors are requested to shed light on the same.

Author's Response to Decision Letter for (RSOS-210790.R1)

See Appendix E.

RSOS-210790.R2

Review form: Reviewer 6

Is the manuscript scientifically sound in its present form?

Yes

Are the interpretations and conclusions justified by the results?

Yes

Is the language acceptable?

Yes

Do you have any ethical concerns with this paper?

No

Have you any concerns about statistical analyses in this paper?

No

Recommendation?

Accept with minor revision (please list in comments)

Comments to the Author(s)

The authors could respond to all the queries. The concern about PFDTS sol-gel is well addressed in the Authors' response section. However, in the actual manuscript, lines 335-337, the statements are creating confusion. (Refer to the statement "Therefore, the antibiosis...." The relevance of this line and the earlier line is not really clear.) Rather authors can use the same justification provided in the Authors' response section, at suitable places.

With this minor change, the manuscript can be accepted for publication.

Decision letter (RSOS-210790.R2)

Dear Dr Li:

Title: Moisture- and Mold-proof Characteristics of Surface Modified Wood for Musical Instrument Soundboards

Manuscript ID: RSOS-210790.R2

It is a pleasure to accept your manuscript in its current form for publication in Royal Society Open Science. The chemistry content of Royal Society Open Science is published in collaboration with the Royal Society of Chemistry.

The comments of the reviewer(s) who reviewed your manuscript are included at the end of this email. Please do consider these when submitting your proof corrections.

Yours sincerely,

Dr Ellis Wilde

Publishing Editor, Journals

On behalf of the Subject Editor Professor Anthony Stace and the Associate Editor Dr Dattatray Late.

RSC Associate Editor: 1
Comments to the Author:
Accept with minor revisions

RSC Associate Editor: 2
Comments to the Author:
(There are no comments.)

Reviewer(s)' Comments to Author:

Reviewer: 6

Comments to the Author(s)

The authors could respond to all the queries. The concern about PFDTS sol-gel is well addressed in the Authors' response section. However, in the actual manuscript, lines 335-337, the statements are creating confusion. (Refer to the statement "Therefore, the antibiosis...." The relevance of this line and the earlier line is not really clear.) Rather authors can use the same justification provided in the Authors' response section, at suitable places.

With this minor change, the manuscript can be accepted for publication.

Appendix A

The manuscript entitled “Moisture- and Mold-proof Characteristics of Surface Modified Wood for Musical Instrument Soundboards” by Xu, Caiping *et al* deals with coating treatment of wood species genus *Paulownia elonata* with solution of polyurethane (PU) and a few w% of MgAl layered double hydroxide (MgAl-LDH) and the subsequent characterization of the treated wood with contact angle (CA) measurements, Fourier Transform infrared (FTIR) spectroscopy, SEM imaging, energy dispersive spectroscopy (EDS) and elemental mapping. The treated wood was further characterized for mold-proofing and for acoustic performance. The manuscript presents a novel wood treatment approach which is topical and contributes to the knowledge in the field of wood treatment and applications. However, the manuscript needs substantial improvement in several aspects. Firstly, the experimental section needs clarification and many experimental details are missing. Secondly, results and discussion section lacks comparison with results obtained by other methods (e.g. thermal treatment, chemical surface modification and plasma treatment). For example, the LDH modified wood exhibited hydrophobicity, how does that compare to results obtained with other methods? Also, the LDH nanosheets have been incorporated on to wood in an earlier study (Reference 14 in this manuscript. <https://doi.org/10.1021/acsami.7b06803>) which can be used to compare EDS/elemental mapping and the incorporation of LDH nanosheets onto wood. Thirdly, wood is natural construction material which has an inhomogeneous and anisotropic structure. This causes deviations in the measured values (e.g. liquid contact angle) caused by wood structure. To account for this, studies with wood typically involve many (upwards from 10) samples/test points for each testing condition. A minimum 3-5 samples should be analyzed for each test condition, but the authors of this manuscript in most cases do not mention anything about how many test points were measured which is a concern. In the manuscript it should be indicated at each experiment how many samples were tested and how repeatable they were. Emphasis should also be placed on determining whether observed trends are statistically relevant with regards to measurement uncertainties. Lastly, the use of English language is severely lacking, and a language corrector/editor is strongly advised.

The following are a list of specific suggestions, comments and questions about the manuscript that need addressing.

Abstract

The abstract in the current state is quite poor. Please rewrite the abstract so that the context of the work and the findings of this work are clearly indicated. The reader should be able to understand most of the contents of the work from the abstract.

Introduction

Introduction need substantial improvement and addition of references to relevant works on thermal wood treatment and in the context of environmentally friendly technologies for wood treatment, also plasma treatment of wood should be mentioned (e.g. works of A. Wolkenhauer, D. Altgen, G. Avramidis *et al*).

Page 3 lines 21-23 The sentence starting with “This vulnerability of wood...” please rephrase the sentence or omit altogether.

In the second paragraph of the introduction (on wood treatment) there should be at least mention of plasma treatment of wood developed over several decades as a green approach to wood treatment (e.g. works of A. Wolkenhauer, D. Altgen, G. Avramidis *et al* or more recent works of R. Talviste, O. Galmiz *et al*).

Page 3 lines 31-32 “...very eco-friendly...” and “...pretreatment...” Please specify what is meant by this and note that thermal treatment is primarily/also used as a separate treatment. Please include references to relevant works. The authors can refer to the review on heat treatment (B. Esteves and H. Pereira “Wood modification by heat treatment: a review, *BioResources* 2009) and the references therein.

Page 3 lines 35-37 The sentence “....., yet the protection effect of this method requires being strengthened.” Please specify what is meant by this (i.e. what are the shortcoming of thermal treatment) and add generally accepted key references on thermal modification of wood (e.g. works by the group of H. Militz, M. Altgen *et al*).

Page 3 lines 35-37 The statement “Additionally, the strength of wood will decrease depending on conditions of thermal treatment ” requires a proper reference.

Page 3 lines 46-48 The sentence “.....for the modification of musical instrument soundboard” should be clarified to e.g “...for the modification of wood suitable for application as musical instrument soundboard.”

Page 3 lines 48-50 The beginning of sentence “Fortunately, the thriving of nanoscience....” Please rephrase to a more neutral wording taking into account the statements by Papadopoulos *et al* in table 1 in “Nanomaterials and chemical modification for enhanced key wood properties: a review” (*Nanomaterials* 9, 2019) on potential hazards of nanomaterials.

Last paragraph of the introduction should be improved to better reflect the motivation of the work and clearly state the aims of the work.

Page 4 lines 21-22 What exactly is meant by “environmentally friendly”? What properties characterized in references 14 and 15 do the authors of this manuscript suggest to be “environmentally friendly”?

Experiment

At the present this section is confusing, unclear and a lot of experimental details is missing. Furthermore, a lot of experimental details are scattered around in the manuscript under each different sub-heading in the “results and discussion” section and should be gathered here in the “experiment” section”. All experimental information should be in this section not in the later sections (e.g, in section “acoustic properties” a substantial part of experiment is described).

First suggestion is to change the name of this section to “Experiment and methods” because several different characterization methods were employed and are described in this section. It is not clear in the experiment section which characterization method is used for MgAl-LDH characterization and which is used for treated wood characterization. This is very confusing to the reader. Many experimental details are missing and should be included in the manuscript for the used characterization methods:

SEM- What detector was used? Were the samples coated before imaging? Were any precautions taken to avoid charging of wood sample surface during imaging? What accelerating voltage was used?

XRD- How were the spectra calibrated and processed? Were the spectra referenced? What was the pass energy and resolution? What was the power of the X-ray beam and the beam spot size?

Contact angle measurements- What wood surface were the measurements carried out on? (e.g. longitudinal, tangential or transverse?) What liquid was used for contact angle measurements? Why were only 5 contact angle measurements carried out for each sample? In many works describing contact angle measurements of wood surfaces minimum of 20-30 measurements are carried out and an average contact angle of those 20-30 measurements is used.

FTIR- What was the resolution? What position on the samples was used for FTIR measurements? How many points/samples were measured? Were the FTIR signals reproducible?

EDS and elemental mapping- please include details of these methods. What is in analysis depth of these methods? Does elemental mapping allow to calculate a concentration/amount of LDH on the surface? (if yes please include in the text) How reproducible were the measurements? What is the measurement uncertainty of these methods?

Mold analysis- Please list the experimental details in the experiment section and not in the “results and discussion” section.

Acoustic properties- All the experimental details listed in the “results and discussion” section should be in the experiment section instead.

Results and discussion

Characterization of modified wood

Page 7 lines 36-45 A list of experimental details that belong in the “experiment” section of the manuscript.

Page 8 lines 37-38 How were the pore sizes determined? Please include in the “experiment” section.

Page 8 lines 41-43 The sentence “However, due to the PU coating, most surface holes on the modified wood presented blocked appearance.” is not clear, please rephrase. Also, please indicate on SEM images where this effect is seen.

Page 8 lines 45-51 Were the LDH nanosheets also visible on the longitudinal surface (figure S3)? If yes/no please indicate also in text and if no give explanation.

Page 8 lines 51-55 Please indicate why these figures provide evidence of LDHs on wood (i.e. the appearance of Mg and Al). Is it possible to estimate quantitatively the amount on LDH nanosheets on the surface from EDS/elemental mapping? How many samples per each condition were investigated with EDS? What was the statistical deviation? Was elemental mapping only done for the transverse section? How do the EDS/elemental mapping results of this work compare to those described in <https://doi.org/10.1021/acsami.7b06803>? For the clarity of this work it might be beneficial to show either EDS or elemental mapping results (either S4 or S5) in the manuscript (instead of supplementary material).

Page 8 lines 55-60 and page 9 line 3 How many samples and points on each sample were investigated with EDS? Please indicate measurement uncertainty and verify that the differences in Mg and Al concentration between longitudinal and transverse sections (and the statement on lines 55-60) are statistically relevant (with proper statistical analysis).

Page 9 lines 4-12 It is unclear what is being measured on figure S6b. Please indicate the different wood cell elements on the SEM figure and describe how the authors determined that the structure being measured represents the thickness of the coating. For the reviewer it is not clear from figure S6 that the measured structure represents the thickness and not just the width of cracked coating.

Figure 3 Images 3a, 3b, 3c, 3d and 3e show the same structure and because there are no difference in those images they are redundant. It is sufficient to show only control and 1 treatment condition (e.g. 2% of LDH in PU) and move the others images (3b, 3c and 3e) to the supplementary material and indicate in text that no visible changes occurred for other conditions. Instead it is worth considering putting on figure 3 images of the longitudinal direction (e.g. control, 2% of LDH in PU and magnification of nanosheets) from the supplementary material.

Properties of modified wood

Since the authors claim that the LDH modified wood is more suitable for soundboards of musical instruments they should compare their results with results from heat treatment, chemical treatment and also plasma treatment of wood. Wood can be made more hydrophobic (CA increases and wetting is reduced) with other treatments e.g. thermal treatment, chemical modification but also plasma treatment as found recently. In particular plasma treatment also only affect the surface and the other key bulk properties of wood are retained (<https://doi.org/10.1007/s00226-020-01175-4>). One of the key experiments, the mold proof test lacks any kind of comparison with existing literature. How does the mold-proof characteristics obtained by LDH nanosheet modification compare to other methods?

Page 9 lines 54-56 The sentence “The rich amount of -OH on the surface of LDH endowed its hydrophilicity.” has no meaning in the context of PU coating in the previous sentence. Please rephrase.

Page 9 lines 56-60, page 10 lines 4-12 and figure 4 The argumentation about the contact angle values is not correct. The argued increase of contact angle from PU (CA of 92°) to 0.5% (CA of 96°) and 1% (CA of 98°) of LDH in PU is less than the uncertainty of the measurements which according to the data provided from 5 measurements is at minimum (taken equal to +/- the standard deviation) +/- 4.5° but about +/- 7° for PU samples. Therefore, the CA values of PU and 0.5%/1% LDH in PU coincide within uncertainty. A comment from the authors is required on how the uncertainty was found on figure 4. The trend of reduction of the contact angle with w% of LDH content above 2% seems to be bear meaning. However, a more reliable result would be obtained from measuring CA-s of about 20-30 droplets for each condition to verify the presently shown decrease of contact angle with increasing LDH content.

page 10 lines 36-42 The definition of leeching rate and relevance to retention level should be moved to the “experiment” section.

page 10 lines 36-42 This statement requires a reference.

page 10 lines 54-56 This statement has not been proven in this work, it is a suggestion of why the leeching rate of LDH containing coating is less, so please rephrase to reflect that. (e.g. The possible reason for decreased leeching of coating containing LDH is due to....)

page 10 lines 55-56 The phrase “While too much” is not correct language for scientific publications, please rephrase. (e.g. The increased w% of LDH in the PU coating can decrease....)

page 11 lines 4-6 The sentences about testing of water absorption rates should be moved to the “experiment” section.

page 11 lines 11-12 What is meant by “adsorption rate of 80%”? Rate is a quantity per unit of time. Is it meant “the water uptake of 80%”?

page 11 lines 17-18 Same question: What is meant by “adsorption rate of 32%”?

page 11 lines 19-21 Water uptake of 32% is not exactly “waterproof performance” please rephrase. (e.g. absorbs water better/worse). Phrase “which probably due to the enhanced hydrophobicity” needs rephrasing.

Figure 5 The authors should indicate how many test samples were investigated (in case of wood it should be minimum 3-5) and what was the uncertainty of the water uptake % measurements. The results shown on figure 5 actually support the trend of decreasing contact angle at LDH w% higher than 1% shown on figure 4. This should also be indicated in the text.

page 11 lines 45-55 Listed here are experimental details that should be moved to the “experiment” part of the work.

Figure 5 The authors should indicate how many samples were investigated for each condition and how was the measurement uncertainty found.

page 12 lines 45-60 and page 13 lines 4-16 A thorough description of the acoustic experiments that were performed that should be moved to the “experiment” section.

Table 1 The authors should indicate how many samples were investigated for each condition and what was the measurement uncertainty. Are the indicated changes after LDH incorporation relevant or not? i.e. did the acoustic properties change slightly or not at all?

Figure 7 Can you please explain why the two peaks of PU/LDH-1 and PU/LDH-2 in the higher frequency region (above 3500 Hz) are shifted to higher frequencies as compared to the control and PU samples which’ peaks coincide in the whole frequency range.

Conclusions

Conclusions in the present state are quite poor, the authors can refer to e.g. ref. 14 (<https://doi.org/10.1021/acsami.7b06803>) for how to write a better summary/conclusion. Conclusions and the entire text direly need language editing by a professional.

Appendix B

Dear editor,

We have substantially revised our manuscript after reading the comments provided by the reviewer 1. The revised sentences have been colored by yellow in the latest manuscript.

Point by point response to the reviewer 1:

The manuscript entitled “Moisture- and Mold-proof Characteristics of Surface Modified Wood for Musical Instrument Soundboards” by Xu, Caiping *et al* deals with coating treatment of wood species genus *Paulownia elonata* with solution of polyurethane (PU) and a few w% of MgAl layered double hydroxide (MgAl-LDH) and the subsequent characterization of the treated wood with contact angle (CA) measurements, Fourier Transform infrared (FTIR) spectroscopy, SEM imaging, energy dispersive spectroscopy (EDS) and elemental mapping. The treated wood was further characterized for mold-proofing and for acoustic performance. The manuscript presents a novel wood treatment approach which is topical and contributes to the knowledge in the field of wood treatment and applications. However, the manuscript needs substantial improvement in several aspects. Firstly, the experimental section needs clarification and many experimental details are missing. Secondly, results and discussion section lacks comparison with results obtained by other methods (e.g. thermal treatment, chemical surface modification and plasma treatment). For example, the LDH modified wood exhibited hydrophobicity, how does that compare to results obtained with other methods? Also, the LDH nanosheets have been incorporated on to wood in an earlier study (Reference 14 in this manuscript. <https://doi.org/10.1021/acsami.7b06803>) which can be used to compare EDS/elemental mapping and the incorporation of LDH nanosheets onto wood. Thirdly, wood is natural construction material which has an inhomogeneous and anisotropic structure. This causes deviations in the measured values (e.g. liquid contact angle) caused by wood structure. To account for this, studies with wood typically involve many (upwards from 10) samples/test points for each testing condition. A minimum 3-5 samples should be analyzed for each test condition, but the authors of this manuscript in most cases do not mention anything about how many test points were measured which is a concern. In the manuscript it should be indicated at each experiment how many samples were tested and how repeatable they were. Emphasis should also be placed on determining whether observed trends are statistically relevant with regards to measurement uncertainties. Lastly, the use of English language is severely lacking, and a language corrector/editor is strongly advised. The following are a list of specific suggestions, comments and questions about the manuscript that need addressing.

At first, thank you very much for your careful read of our manuscript. We have taken your constructive comments into serious consideration and revise the manuscript accordingly. The revised sentences have been colored by yellow in the latest manuscript. For several main issues pointed at the beginning, the relevant revisions were made as the list below:

- 1) The experimental section has been reorganized to 6 parts, namely, the synthesis of LDH, the modification of wood, material characterizations, moisture repellency tests, mold repellency tests and acoustic property tests. Thus, the experimental section would be more logically and clearly. Several experimental details have been added.
- 2) The heat treatment and sol-gel process have been applied to parallelly modify the wood samples in order to conduct comparison with our modification method.
- 3) We have repeated most of our measurements and the deviation values were appended for those data type experimental results.
- 4) The language of the manuscript has been polished for better reading.

Abstract

The abstract in the current state is quite poor. Please rewrite the abstract so that the context of the work and the findings of this work are clearly indicated. The reader should be able to understand most of the contents of the work from the abstract.

Thank you for your suggestion, we have rewritten the abstract to make it more informative and indicative. For your quick reviewing, the present edition is also shown here:

Wood is the major material for musical instrument soundboards fabrication on account of practical and cultural reasons. However, as a natural material, wood is easy to be degraded due to moisture or fungi corrosion. The traditional wood protection methods are normally meant for the structural materials, which might not suitable for the soundboard materials. Therefore, in this work, a novel nanomaterial-based modification method was applied on wood. The surface of wood was coated by polyurethane (PU) and MgAl layered double hydroxide (MgAl-LDH) nanosheets after a convenient impregnation process. The modified wood had improved hydrophobicity and mold-resistance while maintained the acoustic properties, which could do a better work in constructing soundboards with long lifespan.

Introduction

Introduction need substantial improvement and addition of references to relevant works on thermal wood treatment and in the context of environmentally friendly technologies for wood treatment, also plasma treatment of wood should be mentioned (e.g. works of A. Wolkenhauer, D. Altgen, G. Avramidis *et al*).

Thank you for your advice, the mentioned references have been added in the manuscript (Reference 7, 8).

Page 3 lines 21-23 The sentence starting with “This vulnerability of wood...” please rephrase the sentence or omit altogether.

Thank you for your advice, this sentence has been deleted and integrated the main meaning to the next sentence.

In the second paragraph of the introduction (on wood treatment) there should be at least mention of plasma treatment of wood developed over several decades as a green approach to wood treatment (e.g. works of A. Wolkenhauer, D. Altgen, G. Avramidis *et al* or more recent works of R. Talviste, O. Galmiz *et al*).

Thank you for your advice, we have added the plasma treatment into the introduction, and the relevant refers are Reference 7, 11, 12, 13.

Page 3 lines 31-32 "...very eco-friendly..." and "...pretreatment..." Please specify what is meant by this and note that thermal treatment is primarily/also used as a separate treatment. Please include references to relevant works. The authors can refer to the review on heat treatment (B. Esteves and H. Pereira "Wood modification by heat treatment: a review, *BioResources* 2009) and the references therein.

Page 3 lines 35-37 The sentence "... , yet the protection effect of this method requires being strengthened." Please specify what is meant by this (i.e. what are the shortcoming of thermal treatment) and add generally accepted key references on thermal modification of wood (e.g. works by the group of H. Militz, M. Altgen *et al*).

Page 3 lines 35-37 The statement "Additionally, the strength of wood will decrease depending on conditions of thermal treatment " requires a proper reference.

Thank you for your advice. The eco-friendly property of thermal treatment has been specified as "... since it would not generate any extra pollutant than the original wood". The word "pretreatment" was replaced by "modification". We previously used this word simply to demonstrate that the thermal treatment was a process step for timbers, which indeed caused some confusion.

The shortcomings of thermal treatment were specified as "The chief drawback of the treatment is that it causes the mechanical properties degradation of the wood. Also, the wood color becomes darker after the thermal treatment."

The suggested references have been added as Reference 8, 10.

Page 3 lines 46-48 The sentence ".....for the modification of musical instrument soundboard" should be clarified to e.g. "...for the modification of wood suitable for application as musical instrument soundboard."

Thank you for your advice. This sentence has been amended as "Therefore, it is still a great challenge to design a green, efficient and gentle strategy for the modification of wood applied for musical instrument soundboards."

Page 3 lines 48-50 The beginning of sentence "Fortunately, the thriving of nanoscience..." Please rephrase to a more neutral wording taking into account the statements by Papadopoulos *et al* in table 1 in "Nanomaterials and chemical modification for enhanced key wood properties: a review" (*Nanomaterials* 9, 2019) on potential hazards of nanomaterials.

Thank you for your advice. The word “thriving” has been replaced by “development”, and the word “attractive” has been replaced by “alternative” for a more neutral wording.

Last paragraph of the introduction should be improved to better reflect the motivation of the work and clearly state the aims of the work.

Thank you for your advice. We have revised the last paragraph of the introduction to emphasize more about the aims of our work.

Page 4 lines 21-22 What exactly is meant by “environmentally friendly”? What properties characterized in references 14 and 15 do the authors of this manuscript suggest to be “environmentally friendly”?

Thank you for your questions. The layered double hydroxides (LDHs) are a class of ionic solids occur in nature as minerals, and they are usually byproducts of metabolism of certain bacteria. Thus, LDHs are naturally “environmentally friendly”, just as the saying goes, ashes to ashes, and dust to dust. After our carefully literature research, there is little measurement result about “environmentally friendly” property of LDHs, while a lot of articles write this characteristic in their titles or introduction parts. So, we added another reference (Reference 23) to the manuscript in which the environmentally friendly property of LDH is emphasized.

Experiment

At the present this section is confusing, unclear and a lot of experimental details is missing. Furthermore, a lot of experimental details are scattered around in the manuscript under each different sub-heading in the “results and discussion” section and should be gathered here in the “experiment” section”. All experimental information should be in this section not in the later sections (e.g, in section “acoustic properties” a substantial part of experiment is described). First suggestion is to change the name of this section to “Experiment and methods” because several different characterization methods were employed and are described in this section. It is not clear in the experiment section which characterization method is used for MgAl-LDH characterization and which is used for treated wood characterization. This is very confusing to the reader. Many experimental details are missing and should be included in the manuscript for the used characterization methods:

Thank you for your comments. We are very sorry for causing your confusion. The section has been changed to “Experiment and methods”. The experimental section has been reorganized to 6 parts, namely, the synthesis of LDH, the modification of wood, material characterizations, moisture repellency tests, mold repellency tests and acoustic property tests. Thus, the experimental section would be more logically and clearly. Several experimental details have been added and the scattered information about the experiments and characterizations in “Results and discussion” section have been integrated into the “Experiment and methods” section.

SEM- What detector was used? Were the samples coated before imaging? Were any precautions taken to avoid charging of wood sample surface during imaging? What accelerating voltage was used?

Thank you for your comments. The SEM images were acquired by the electron backscatter diffraction (EBSD) detector at 10 kV accelerating voltage with a working distance of 10 mm. For the LDH characterization, the samples were directly observed by SEM without extra treatment. For the wood samples, first, all the samples were dried at 100 °C for another 5 h. After that, the samples were sputtered with a thin layer of Au by a vacuum evaporation coating apparatus (ETD-800C, Beijing Boyuan Micro Nano Technology Co. Ltd).

XRD- How were the spectra calibrated and processed? Were the spectra referenced? What was the pass energy and resolution? What was the power of the X-ray beam and the beam spot size?

Thank you for your comments. The calibration of XRD was performed by the engineer once a month using the standard Si sample. The spectra of LDH were referred to PDF card 89-0460, as shown in Figure 1f. The running parameters were kept at 40kV, 35mA. The resolution was 0.02°/s.

Contact angle measurements- What wood surface were the measurements carried out on? (e.g. longitudinal, tangential or transverse?) What liquid was used for contact angle measurements? Why were only 5 contact angle measurements carried out for each sample? In many works describing contact angle measurements of wood surfaces minimum of 20-30 measurements are carried out and an average contact angle of those 20-30 measurements is used.

Thank you for your comments. The longitudinal (3 points) and transverse (2 points) surfaces of the wood samples were taken for the contact angle measurements. The testing liquid was distilled water. We are very sorry for the unprofessional measurements. The contact angle values in the revise manuscript were determined by 20 separate measurements for one sample. To be specific, each wood block had 5 test points and each kind of modified wood had 4 parallel test samples.

FTIR- What was the resolution? What position on the samples was used for FTIR measurements? How many points/samples were measured? Were the FTIR signals reproducible?

Thank you for your comments. The resolution of FTIR is 4 cm⁻¹ for 32 scans in the range from 700 cm⁻¹ to 4000 cm⁻¹. The testing samples were obtained by cutting the surfaces of various wood samples and ground to make small chips. Three different chips from the same wood sample were analyzed in order to ensure that the obtained spectra

represent the whole board. The FTIR signals were reproducible since we recently took the FTIR characterizations again and obtained nearly the same spectra as before.

EDS and elemental mapping- please include details of these methods. What is in analysis depth of these methods? Does elemental mapping allow to calculate a concentration/amount of LDH on the surface? (if yes please include in the text) How reproducible were the measurements? What is the measurement uncertainty of these methods?

Thank you for your comments. The depth of EDS is 100~1000 nm. The EDS characterization was taken by the Oxford Aztec EDX instrument equipped in the SU-70 SEM, the results could only be qualitative or half-quantitative. As a matter of fact, the wood surface is very rough, which has huge influence on the quantitative accuracy of EDS. Also, in our work, the amount of LDH nanosheets on the wood surface is relatively small, adding the difficulty for precise quantification. The relevant study and discussion about the quantification ability of SEM/EDS is done by Dale E. Newbury et al (<https://onlinelibrary.wiley.com/doi/10.1002/sca.21041>). Therefore, it is difficult for the EDS characterization to measure the amount of LDH on the wood surface in our work.

Mold analysis- Please list the experimental details in the experiment section and not in the “results and discussion” section.

Acoustic properties- All the experimental details listed in the “results and discussion” section should be in the experiment section instead.

Thank you for your comments. These descriptions have been moved to the experiment section.

Results and discussion

Characterization of modified wood

Page 7 lines 36-45 A list of experimental details that belong in the “experiment” section of the manuscript.

Thank you for your advice. We have deleted sentences describing experimental details in this section and moved to the “experiment and method” section.

Page 8 lines 37-38 How were the pore sizes determined? Please include in the “experiment” section.

Thank you for your question. The pore sizes were determined by measuring the diameter of different pores in several micrographs of wood. The measurement was done by a software called SmileView.

Page 8 lines 41-43 The sentence “However, due to the PU coating, most surface holes

on the modified wood presented blocked appearance.” is not clear, please rephrase. Also, please indicate on SEM images where this effect is seen.

Thank you for your advice. We have rephrased this sentence to “However, ..., due to the PU coating, the modified wood presented an adhesive layer.” and changed the SEM images to higher magnification ones for clearer observation of the PU existence.

Page 8 lines 45-51 Were the LDH nanosheets also visible on the longitudinal surface (figure S3)? If yes/no please indicate also in text and if no give explanation.

Thank you for your comments. In the previous manuscript, the magnification of SEM images is not high enough. Therefore, we repeated the SEM observation of the samples and obtained clearer images. The LDH nanosheets could also be visible on the longitudinal surface, as shown in the Figure 3d₃. For your quick reviewing, the image is also put below:

Page 8 lines 51-55 Please indicate why these figures provide evidence of LDHs on wood (i.e. the appearance of Mg and Al). Is it possible to estimate quantitatively the amount on LDH nanosheets on the surface from EDS/elemental mapping? How many samples per each condition were investigated with EDS? What was the statistical deviation? Was elemental mapping only done for the transverse section? How do the EDS/elemental mapping results of this work compare to those described in <https://doi.org/10.1021/acsami.7b06803>? For the clarity of this work it might be beneficial to show either EDS or elemental mapping results (either S4 or S5) in the manuscript (instead of supplementary material).

Page 8 lines 55-60 and page 9 line 3 How many samples and points on each sample were investigated with EDS? Please indicate measurement uncertainty and verify that the differences in Mg and Al concentration between longitudinal and transverse sections (and the statement on lines 55-60) are statistically relevant (with proper statistical analysis).

Since the EDS characterization was taken by the Oxford Aztec EDX instrument equipped in the SU-70 SEM, the results could only be qualitative or half-quantitative. Moreover, the LDH content in our work was far less than that in the work you

mentioned (<https://doi.org/10.1021/acsami.7b06803>). In their work, the LDH nanosheets densely grow on the wood surface by a sol-gel processing and constitute the protecting layer. The wood surface is very rough, which has huge influence on the quantitative accuracy of EDS. Therefore, it is difficult to use the EDS characterization to calculate the exact amount of LDH on the wood surface. The relevant study and discussion about the quantification ability of SEM/EDS is done by Dale E. Newbury et al (<https://onlinelibrary.wiley.com/doi/10.1002/sca.21041>). We have also done the elemental mapping for the transverse section, which is shown in Figure S5. After repeated EDS characterizations for the longitudinal and transverse surfaces of the samples, we found the result deviation was large (see the Figures below, which practically proves that the SEM/EDS is not suitable for quantification). Hence, the statement about the LDH amount difference between these two kinds of surfaces was deleted.

Page 9 lines 4-12 It is unclear what is being measured on figure S6b. Please indicate the different wood cell elements on the SEM figure and describe how the authors determined that the structure being measured represents the thickness of the coating. For the reviewer it is not clear from figure S6 that the measured structure represents the thickness and not just the width of cracked coating.

Thank you for your comments. After careful consideration and discussion between co-workers, we decide to delete the previous Figure S6. It is truly hard to say that the labeled region in Figure S6b could represent the coating layer. The further repeated SEM observation also did not get trustful evidence.

Figure 3 Images 3a, 3b, 3c, 3d and 3e show the same structure and because there are no difference in those images they are redundant. It is sufficient to show only control and 1 treatment condition (e.g. 2% of LDH in PU) and move the others images (3b, 3c and 3e) to the supplementary material and indicate in text that no visible changes occurred for other conditions. Instead it is worth considering putting on figure 3 images of the longitudinal direction (e.g. control, 2% of LDH in PU and magnification of nanosheets) from the supplementary material.

Thank you for your advice. We have reorganized all the morphology characterization results. Instead of putting all the results to make readers feel redundant about most of the images, we selected representative images to show the microscopic morphological observation. The surfaces of pure wood, PU coated wood and PU/LDH-6 were mainly discussed.

Properties of modified wood

Since the authors claim that the LDH modified wood is more suitable for soundboards of musical instruments they should compare their results with results from heat treatment, chemical treatment and also plasma treatment of wood. Wood can be made more hydrophobic (CA increases and wetting is reduced) with other treatments e.g.

thermal treatment, chemical modification but also plasma treatment as found recently. In particular plasma treatment also only affect the surface and the other key bulk properties of wood are retained (<https://doi.org/10.1007/s00226-020-01175-4>). One of the key experiments, the mold proof test lacks any kind of comparison with existing literature. How does the mold-proof characteristics obtained by LDH nanosheet modification compare to other methods?

Thank you for your comments. We have applied simplified heat treatment and sol-gel process to modify the wood to make comparison. As for the plasma treatment method, after careful reading the related articles, we found that our lab did not have the suitable instruments to fulfil this modification. Though we have tried to use the plasma cleaning machine in our lab, the treated wood samples did not become hydrophobic.

Page 9 lines 54-56 The sentence “The rich amount of -OH on the surface of LDH endowed its hydrophilicity.” has no meaning in the context of PU coating in the previous sentence. Please rephrase.

Thank you for your comments. We have rephrased this sentence to “Normally, the rich amount of -OH on the surface of LDH is supposed to endow its hydrophilicity.” Therefore, this sentence would play the role to draw forth the phenomenon described in the next sentence.

Page 9 lines 56-60, page 10 lines 4-12 and figure 4 The argumentation about the contact angle values is not correct. The argued increase of contact angle from PU (CA of 92°) to 0.5% (CA of 96°) and 1% (CA of 98°) of LDH in PU is less than the uncertainty of the measurements which according to the data provided from 5 measurements is at minimum (taken equal to +/- the standard deviation) +/- 4.5° but about +/- 7° for PU samples. Therefore, the CA values of PU and 0.5%/1% LDH in PU coincide within uncertainty. A comment from the authors is required on how the uncertainty was found on figure 4. The trend of reduction of the contact angle with w% of LDH content above 2% seems to be bear meaning. However, a more reliable result would be obtained from measuring CA-s of about 20-30 droplets for each condition to verify the presently shown decrease of contact angle with increasing LDH content.

Thank you for your comments. We repeated the contact angle measurements and the CA values in the revised manuscript were determined by 20 separate measurements for one sample. As shown in Figure 4, the tendency does not change: as the amount of LDH doping increases, the contact angle first increased slightly and then decreased. However, considering the deviation values, the PU/LDH-0.5, PU/LDH-1 and PU/LDH-2 actually had close CA values.

page 10 lines 36-42 The definition of leaching rate and relevance to retention level should be moved to the “experiment” section.

Thank you for your comments. The relevant description has been moved.

page 10 lines 36-42 This statement requires a reference.

Thank you for your advice. The Reference 19 has been attached to this sentence.

page 10 lines 54-56 This statement has not been proven in this work, it is a suggestion of why the leeching rate of LDH containing coating is less, so please rephrase to reflect that. (e.g. The possible reason for decreased leeching of coating containing LDH is due to....)

page 10 lines 55-56 The phrase “While too much” is not correct language for scientific publications, please rephrase. (e.g. The increased w% of LDH in the PU coating can decrease....)

Thank you for your advice. These two sentences have been rephrased.

page 11 lines 4-6 The sentences about testing of water absorption rates should be moved to the “experiment” section.

Thank you for your advice. The relevant descriptions have been moved to the experiment section.

page 11 lines 11-12 What is meant by “adsorption rate of 80%”? Rate is a quantity per unit of time. Is it meant “the water uptake of 80%”?

page 11 lines 17-18 Same question: What is meant by “adsorption rate of 32%”?

page 11 lines 19-21 Water uptake of 32% is not exactly “waterproof performance” please rephrase. (e.g. absorbs water better/worse). Phrase “which probably due to the enhanced hydrophobicity” needs rephrasing.

Thank you for your comments. The word “rate” has been changed to “value” and the definition of water adsorption (WA) value was written in the experiment section. Namely, the WA values were determined using the following equation.

$$WA (\%) = 100 \times (M_2 - M_1) / M_1$$

where WA is the water absorption, M_1 is the weight of the specimen before tests, and M_2 is the weight of the water-saturated specimen.

Figure 5 The authors should indicate how many test samples were investigated (in case of wood it should be minimum 3-5) and what was the uncertainty of the water uptake % measurements. The results shown on figure 5 actually support the trend of decreasing contact angle at LDH w% higher than 1% shown on figure 4. This should also be indicated in the text.

page 11 lines 45-55 Listed here are experimental details that should be moved to the “experiment” part of the work.

Figure 5 The authors should indicate how many samples were investigated for each condition and how was the measurement uncertainty found.

Thank you for your comments. The data in Figure 5 is shown in Table S2, with deviation included. The test sample number was 3 for each condition. The details about the measurement uncertainty have been included in the experiment section.

page 12 lines 45-60 and page 13 lines 4-16 A thorough description of the acoustic experiments that were performed that should be moved to the “experiment” section.

Table 1 The authors should indicate how many samples were investigated for each condition and what was the measurement uncertainty. Are the indicated changes after LDH incorporation relevant or not? i.e. did the acoustic properties change slightly or not at all?

Thank you for your comments. The description has been moved to the experiment section with more details. The test times and the deviations have shown in the experiment section and Table 1. Considering the deviations, the LDH incorporation did not cause observable change to the acoustic properties of wood.

Figure 7 Can you please explain why the two peaks of PU/LDH-1 and PU/LDH-2 in the higher frequency region (above 3500 Hz) are shifted to higher frequencies as compared to the control and PU samples which' peaks coincide in the whole frequency range.

Thank you for your comments. Normally, the frequency of wood is correlated to the density, water content, shape and structure. This slight shift might be caused by the density change, as shown in Table 1.

Conclusions

Conclusions in the present state are quite poor, the authors can refer to e.g. ref. 14 (<https://doi.org/10.1021/acsami.7b06803>) for how to write a better summary/conclusion. Conclusions and the entire text direly need language editing by a professional.

Thank you for your comments, we have rewritten the “Conclusions” section. For your quick reviewing, the present edition is also shown here:

By applying PU and nanoscale MgAl-LDH in wood modification, the mold-proof and moisture-proof properties of wood were improved efficiently. The optimum LDH content was determined by evaluating the wood samples modified with different PU/LDH ratios. When the LDH content was 1%, the PU/LDH coated wood had a hydrophobic surface with a contact angle of $118.7^{\circ} \pm 7.1^{\circ}$. The mass loss of the PU/LDH wood sample was smaller than the other control samples after the same mold growth process, which verified the enhanced mold-proof characteristic. The vibration test

showed that the PU/LDH coating did not noticeably affect acoustic properties of the wood. In conclusion, this novel protection strategy is very promising in protecting wood materials used for musical instrument soundboards.

Appendix C

Dear Editors,

We have revised the manuscript entitled “Moisture- and mold-proof characteristics of surface modified wood for musical instrument soundboards” (manuscript ID: RSOS-210790) after carefully reading you and the reviewers’ comments. The revised part has been highlighted in the manuscript. A brief introduction about the novelty of our work is also written below.

Wood is the major raw material for soundboards, and sound qualities of instruments made from wood are highly concerned with the wood properties. However, as a natural biomaterial, wood is easily to degrade because of moisture and mold. The traditional wood modification methods, including thermal and chemical treatments, are usually not suitable for the special timbers used for soundboards. The ideal method is supposed to hold the original appearance and mechanical properties of wood, while enhanced the surface resistances towards moisture and mold. As the development of nanoscience and nanotechnology in recent years, protecting wood with nanomaterials has become another alternative choice. MgAl layered double hydroxide (MgAl-LDH) nanosheet is a cheap, chemical stable and environmentally friendly ion clay material. In this work, the waterborne polyurethane (PU) as well as MgAl-LDH are applied to modify the wood surface. On the one hand, the electrostatic interaction between PU and LDH improves the hydrophobicity of the composite. On the other hand, the two-dimensional (2D) nanostructure and hydroxyl-group-rich surface endow LDH with stronger adsorption ability on wood. The novelty of our work is that the PU/LDH could not only effectively enhance the moisture- and mold-proof abilities of wood, but also preserve the aesthetical and acoustical properties of the pristine wood. Therefore, when it comes to the timbers for musical instrument soundboards, this nanomaterial-based wood treatment strategy could do a better work.

Thanks very much for your attention to our paper.

Very sincerely yours,

Xingyun Li

lixingyun@stu.xmu.edu.cn

Appendix D

Reviewer: 3

Comments to the Author(s)

The only suggestion of mine to improve the literatures is adding a comprehensive reference and citation for the equations provided for the acoustic property tests. The authors forgot to cite a reference there!

I suggest:

<https://doi.org/10.1016/B978-0-12-803581-8.01996-2>

Thank you for your kind suggestion. The reference has been added in the revised manuscript as **Ref.26**.

Reviewer: 4

Comments to the Author(s)

The work is not novel and not upto the standards of the journal

Thank you for your comment. The novelty of our work is mainly about applying the two-dimensional (2D) nanomaterial MgAl-LDH to modify the wood used for music instrument soundboards. In order to better show the novelty of our work, the Introduction part was revised to highlight the essential role of LDH nanosheet. MgAl-LDH is one of the ionic clay materials with unique layered structure that could be exfoliated to 2D nanosheet. The 2D structure render LDH nanosheets with large specific area, which could greatly reduce the consumption of the coverage materials and increase the physical shield effect. The well-known properties of MgAl-LDH include low cost, environmentally friendly, great anticorrosion ability and high anion exchange capacity. By changing the anions, the chemical composition of LDH could be flexibly tuned to modify its characteristics. The rich hydroxyl groups on the LDH surface makes it easy to bind to biomaterials. Therefore, the LDH nanosheet is a very promising material for wood functionalization or modification. The co-modification by PU and LDH could effectively improve the moisture- and mold-proof properties of wood. Furthermore, the modification process had little effect on the acoustic parameters of the original wood, guaranteeing the great vibro-mechanical properties of wood as musical instrument soundboards. The corresponding discussion has been added in Page 3 of the revised manuscript.

Appendix E

Reviewer comments to Author:

Reviewer: 4

Comments to the Author(s)

The paper was not improved

Thank you for the comment, and we feel regretful to give you such an impression. In order to improve our manuscript, we have rewritten the second paragraph in Page 3 to give a more specific introduction about the LDH material, showing that the novelty of our work is to apply this 2D material into the protection of wood used for instruments.

Reviewer: 5

Comments to the Author(s)

Moisture-proof is very important for wood used as musical instrument soundboards. In this work, MgAl layered double hydroxide (MgAl-LDH) nanosheet as a cheap, chemical stable and environmentally friendly ion clay material was synthesized on the surfaces of wood. The co-modification by PU and LDH could effectively improve the moisture- and mold-proof properties of wood. Furthermore, the modification process had little effect on the acoustic parameters of the original wood, guaranteeing the great vibromechanical properties of wood as musical instrument soundboards. Since the novelty of this work has been addressed by the authors. The authors have also improved their manuscript according to the reviewers' comments. I would like to recommend its acceptance for publication.

Thank you for your comments. We are very grateful to your positive comments about our manuscript.

Reviewer: 6

Comments to the Author(s)

The use of inappropriate verbs or prepositions may change the meaning of the sentence, hence better get the manuscript rechecked with some expert in English.

Although authors could discuss the conventional wood protection strategies, the limitations of the metal/metal oxide nanoparticles coating seem missing. Authors directly jump to LDH, leaving the link from nanoparticles(0D) to 2D nanomaterials missing.

The Moisture-proof property, as studied with contact angle measurement (Fig. 4) indicate PFDTs sol-gel treatment to be the best. In such a case, it is hard to understand the reason for skipping the PFDTs sol-gel treatment in some of the subsequent studies. Authors are requested to shed light on the same.

Thank you for your comments. The English of the manuscript has been revised by the native English-speaking editor at the professional editing service, and the revised words were highlighted

by yellow color. The main limitations of the metal/metal oxide nanoparticles are the small size induced aggregation problem and the risk of detachment. The descriptions about the limitations have been appended to the second paragraph of the Introduction section, and highlighted by green color.

The PFDTs sol-gel treatment is a commonly used hydrophobic modification method, here in our work, the wood samples modified by this method are used as contrasts during the moisture and mold resistance tests. According to these contrast results, the mold-proof performance is not entirely decided by the moisture resistance ability. More discussion about the PFDTs sol-gel treatment has been added to the manuscript in Page 12 and 15 (highlighted by green color). Though the PFDTs is a good hydrophobic material, it is rather complicated and expensive (*J. Mater. Chem. A*, 2016, 4, 13677-13725). In addition, the dissolution of PFDTs takes organic solvent. Thus, this method is presently not suitable for wood protect. On the other hand, the PU and LDH are both cost-effective and environmentally friendly materials, but the LDH is a powdered material which has retention issue. Therefore, only the retention level (leaching rate and water absorption value) of PU/LDH samples were measured in our work.